

# An automated method for the evaluation of the pointing accuracy of sun-tracking devices

Dietmar J. Baumgartner[1], Werner Pötzi[1], Heinrich Freislich[1], Heinz Strutzmann[1], Astrid M. Veronig[1,2], and Harald E. Rieder[3,2,4]

[1]Kanzelhöhe Observatory for Solar and Environmental Research (KSO), University of Graz, Austria
[2]Institute for Geophysics, Astrophysics and Meteorology/Institute of Physics (IGAM/IP), University of Graz, Austria
[3]Wegener Center for Climate and Global Change (WEGC), University of Graz, Austria
[4]Austrian Polar Research Institute, Vienna, Austria

*Correspondence to*: D. J. Baumgartner (dietmar.baumgartner@uni-graz.at)

**Abstract.** The accuracy of solar radiation measurements (for direct and diffuse radiation) depends significantly on the precision of the operational sun-tracking device. Thus rigid targets for instrument performance and operation have been specified for international monitoring networks, such as e.g., the Baseline Surface Radiation Network (BSRN) operating under the auspices of the World Climate Research Program (WCRP). Sun-tracking devices fulfilling these accuracy requirements are available from various instrument manufacturers, however none of the commercially available systems comprises an automatic accuracy control system, allowing platform operators to independently validate the pointing accuracy of sun-tracking sensors during operation. Here we present KSO-STREAMS (KSO-SunTRackEr Accuracy Monitoring System), a fully automated, system independent and cost-effective method for evaluating the pointing accuracy of sun-tracking devices. We detail the monitoring system setup, its design and specifications and results from its application to the sun-tracking system operated at the Austrian RADiation network (ARAD) site Kanzelhöhe Observatory (KSO). Results from an evaluation campaign from March to June 2015 show that the tracking accuracy of the device operated at KSO lies for the vast majority of observations (99.8%) within BSRN specifications (i.e., 0.1° tracking accuracy). Evaluation of manufacturer specified active tracking accuracies (0.02°), during periods with direct solar radiation exceeding 300 W m$^{-2}$, shows that these are satisfied for 72.9% of observations. Tracking accuracies are highest during clear-sky conditions and on days where prevailing clear-sky conditions are interrupted by frontal movement: in these cases we obtain complete fulfillment of BSRN requirements and 76.4% of observations within manufacturer specified active tracking accuracies. Limitations to tracking surveillance arise during overcast conditions and periods of partial solar limb coverage by clouds. On days with variable cloud-cover 78.1% (99.9%) of observations meet active tracking (BSRN) accuracy requirements while for days with prevailing overcast conditions these numbers reduce to 64.3% (99.5%), respectively.



# 1 Introduction

A precise knowledge of the surface energy budget, which comprises the solar and terrestrial radiation fluxes, is essential for understanding Earth's climate system (e.g., Wild et al., 2015). The surface radiation budget itself is defined by the difference

of the downward and upward components of short and long wave irradiance (e.g., Augustine and Dutton, 2013). To date ground based measurements provide the most reliable information on short and long wave irradiance. They are routinely utilized for retrieval optimization, the evaluation of satellite radiation products (Pinker et al., 2005; Gupta et al., 2004; Wang et al., 2014; Yan et al., 2011) and the evaluation/parameterization of radiative fluxes in global/regional climate models (e.g., Wild et al., 1998; Marty et al., 2003; Donner et al., 2011; Freidenreich and Ramaswamy, 2011) and reanalysis products (e.g.,

Allan, 2000).

Driven by the increasing need for high accuracy surface radiation data for scientific and technical applications, e.g. to enhance the performance of solar photovoltaic plants (e.g., Fontani et al., 2011), national and international radiation monitoring networks have been established over recent decades. The most prominent international radiation monitoring network is the so called Baseline Surface Radiation Network (BSRN) operating under the auspices of the World Climate

Research Program (WCRP) (e.g., Ohmura et al., 1998). BSRN sites are equipped with instruments of highest accuracy. Targets for instrument performance and operation are specified in the BSRN guidelines (McArthur, 2005). Furthermore BSRN guidelines are (closely) adopted by national radiation monitoring networks, such as e.g., ARAD in Austria (Olefs et al., 2016), SACRaM in Switzerland (Wacker et al., 2011), or SURFRAD in the US (Augustine et al., 2005).

BSRN guidelines require the operation of radiation sensors on sun-tracking devices, with specified accuracy, available, in

various designs, from different instrument manufacturers. BSRN guidelines recommend the use of (i) single-axis synchronous motor tracking devices, (ii) dual axis passive (algorithm controlled) tracking devices or (iii) dual axis active (quadrant sensor controlled) tracking devices. For a detailed overview about advantages and disadvantages of these tracking devices we refer the interested reader to Sect. 4 in McArthur (2005).

Among the suite of solar radiation measurements particularly the accuracy of pyrheliometer (direct solar radiation; DIR) and

pyranometer measurements (diffuse solar radiation; DIF) depends significantly on the accuracy of the operational sun-tracking device. Thus BSRN guidelines recommend using a sun-tracking device with an accuracy of ±0.1° or better to accommodate pyrheliometers and that the tracking is monitored using a four-quadrant sensor as the pointing accuracy is important in determining the quality of the direct beam measurement (McArthur, 2005).

Sun-tracking devices fulfilling these BSRN recommendations are available from various instrument manufacturers.

Nevertheless, none of the commercially available platforms comprises an automatic accuracy control system, allowing platform operators to check if the operational pointing accuracy of the sun-tracking device indeed fulfills BSRN targets. Here we bridge this gap by presenting a fully automated, system independent and cost effective observing system to





determine the pointing accuracy of sun-tracking devices which can be easily added to existing monitoring platforms, and its application to the sun-tracking device operated at Kanzelhöhe Observatory (KSO), Austria. We note that KSO-STREAMS is solely intended to monitor tracking accuracy and not to adjust the alignment of an operational sun-tracking device.

## 2 The proposed observing system for the evaluation of sun-tracking device pointing accuracy

### 2.1 Components and installation

The proposed observing system for continuous monitoring of the alignment (i.e., pointing accuracy) of the sun-tracking device, hereinafter referred to as KSO-STREAMS (KSO-SunTRackEr Accuracy Monitoring System) consists of five key components: (i) a circular aperture, (ii) an optical filter block, (iii) an achromatic lens (fixed focal length of 60 mm), (iv) an adapted compact network camera with corresponding web-connectivity, and (v) a fitted housing and mounting system. The observing system and a schematic illustration of the system components are shown in Figs. 1a and 1b. Details on system components are provided in Table 1. During operation KSO-STREAMS needs to be mounted like a pyrheliometer on the sun-tracking device (Fig. 1c) to ensure correct imaging of the Sun's position as identified by the tracker (computed and adjusted in the case of a four-quadrant sensor correction). The focal length of KSO-STREAMS is chosen to allow the registration of a misalignment of the imaged solar disk of 0.5 degrees (corresponding to approximately two solar radii) in each direction from the image center (see Fig. 2a). This enables us to identify the pointing accuracy of the sun-tracking device and to quantify potential misalignments.

### 2.2 Principle and defined accuracy limits

KSO-STREAMS is operated by an automated script and takes, between sunrise and sunset, every 15 seconds a snapshot of the solar disk. The images taken are immediately processed as detailed below. A typical image taken by KSO-STREAMS is given in Fig. 2b. We note that not image type (i.e. color) but high image contrast and a minimal stray light are important in further steps for solar limb detection. Thus all KSO-STREAMS pictures are first converted to greyscale (see Fig. 3a) and derotated, as it is not possible to mount KSO-STREAMS in perfect horizontal alignment on the sun-tracking system image rotation is necessary during post-processing. To determine the amount of image rotation necessary to achieve the horizontal alignment we follow a four step procedure: (i) the sun-tracking device is positioned and fixed to its local noon position three minutes before actual local noon; (ii) images are recorded in five seconds intervals while the Sun is moving across the whole image plane; (iii) the center of the solar disk is determined (see method described below); (iv) a line is fitted through the solar disk centers recorded and the angle ($\varphi$) from this line to the image border is calculated. Furthermore every image has to be derotated by $\varphi$ to achieve horizontal image alignment.

For each image the solar disk center ($x$-, $y$-position) has to be determined prior to further processing. To this aim we apply a standard Sobel operator (Jaehne, 1991) to the high contrast images obtained, in order to detect the solar limb. The Sobel

operator calculates the image gradient of each pixel by convolving the image with a pair of 3x3 filters, which estimate the gradients in the horizontal and vertical directions. The sum of the gradients in the horizontal and vertical directions yields the magnitude of the gradient. Given the high contrast between Sun and background in KSO-STREAMS images the solar limb is in first order defined through the pixels with the largest gradient (see Fig. 3b).

Next we apply a classical least square circle fit (Ludwig, 1969) to the solar limb pixels identified to calculate the radius and the center of the solar disk. Each of the first order solar limb pixels identified is characterized through coordinates in the $xy$-plane and the classical least square fitting approach minimizes the geometric distance of these points to the fitted circle. The best fit through the set of $n$ points (i.e. the first order identified solar limb pixels) is achieved through minimization of Eq. (1) by solving the system for $\frac{\partial F}{\partial h} = 0$, $\frac{\partial F}{\partial k} = 0$ and $\frac{\partial F}{\partial r} = 0$,

$$F(h,k,r) = \sum_{i=1}^{n}\left[\sqrt{(x_i - h)^2 - (y_i - k)^2} - r^2\right] \rightarrow \min \tag{1}$$

where $(x_i, y_i)$ denote the first order solar limb pixels, and $(h, k)$ the circle center and $r$ the radius of the fitting circle, respectively. As only coordinate pairs $(x_i, y_i)$ are known, the circle equation has to be linearized to obtain a series of linear 15 equations yielding Eq. (4), which is linear in the undetermined coefficients $a$, $b$ and $c$, that allows once $a$, $b$ and $c$ are derived to solve backwards for $h$, $k$ and $r$:

$$r^2 = (x - h)^2 + (y - k)^2 = x^2 - 2hx + h^2 + y^2 - 2ky + k^2 \tag{2}$$

$$x^2 + y^2 = 2hx + 2ky + r^2 - h^2 - k^2 \tag{3}$$

$$x^2 + y^2 = ax + by + c \tag{4}$$

The coefficients $a$, $b$ and $c$ are derived applying the matrix equation for a circular regression (Eq. 5),

$$\begin{bmatrix} \sum x_i^2 & \sum x_i y_i & \sum x_i \\ \sum x_i y_i & \sum y_i^2 & \sum y_i \\ \sum x_i & \sum y_i & n \end{bmatrix} \begin{bmatrix} a \\ b \\ c \end{bmatrix} = \begin{bmatrix} \sum x_i(x_i^2 + y_i^2) \\ \sum y_i(x_i^2 + y_i^2) \\ \sum x_i^2 + y_i^2 \end{bmatrix} \tag{5}$$


where $n$ denotes the number of individual points $(x_i, y_i)$. It follows that a unique set of values for the coefficients $a$, $b$ and $c$ – that generate the circle of best fit – exists, when the 3-by-3 matrix on the left side of Eq. (5) is invertible. After deriving the coefficients $a$, $b$, and $c$ the center $(h, k)$ and radius $r$ can be compute through the following transformations,

$$h = -a/2 \tag{6}$$

$$k = -b/2 \tag{7}$$

$$r = \frac{\sqrt{4c + a^2 + b^2}}{2} \tag{8}$$





and the best fitting circle can be established (see Fig. 3c).

The error on the circle fit is demanded to be less than one pixel. As the solar radius varies throughout the year uncertainty limits for the detection of the solar limb have been defined as +5% of the largest and -5% of the smallest astronomically

calculated solar radius throughout the year. As we use a prime lens (i.e., lens with a fixed focal length of 60 mm) KSO-STREAMS focal length is not an issue. The results of the processing algorithm are stored in daily look up tables for further post processing, and archived to allow retrospective investigation of the representativeness of solar radiation measurements found 'dubious' in further analyses.

If both accuracy conditions are fulfilled an image is considered valid and used for further analysis. It is obvious that turbidity

and cloudiness (and here especially broken cloud coverage in front of the sun) complicate/compromise solar limb detection. This is further investigated in Sect. 3.2 were we analyze sun-tracker pointing accuracies over a wide range of cloud cover conditions ranging from clear sky to perpetual overcast conditions.

## 2.3 Initial zero point determination of the solar center determined by KSO-STREAMS

For initial zero point determination we utilize data from 12 days in mid-March to mid-June 2015 (three days each in March, April, May, June; see Fig. 4) with high (and continuous) availability of observational data (i.e. perpetual clear-sky conditions). Furthermore we restrict the accuracy of limb detection to less than half a pixel. For each day we determine the mean zero point as average over all available solar disk center positions. The initial zero point is then determined as the average of the 12 individual daily mean zero center positions. We note that the difference among individual daily mean zero

positions (in both azimuthal and zenithal direction) is small, i.e. less than 8 Pixels (which corresponds to approximately 0.03° or 6% of the solar disk diameter), and is mainly affected by different atmospheric conditions. In the following the accuracy of the sun-tracking device is characterized by the difference of KSO-STREAMS solar disk centers to this initial zero point center defined.

## 3. Application to the sun-tracking device at Kanzelhöhe Observatory, Austria

### 3.1 Field Measurements

KSO-STREAMS was installed on 12 March 2015 at the sun-tracking device (type: SOLYS 2, Kipp & Zonen) of the Austrian RADiation network (ARAD, Olefs et al. (2016)) station Kanzelhöhe (1540 m a.s.l.), see Fig. 1d. The sun-tracking device is equipped with a sun sensor which allows fine tuning of the alignment to the Sun, if direct radiation is at least 300 W m$^{-2}$. The information from the sun sensor is updated every 10 seconds by the sun-tracking device thus

KSO-STREAMS is operated with 15 second 'snapshots'. Continuous operation (during ARAD site operation) started on 13 March 2015. Below we detail the analysis of the sun-tracking device performance/accuracy as monitored by KSO-STREAMS for 15 weeks during March to June 2015.





## 3.2 Evaluation of the pointing accuracies of sun-tracking devices under different meteorological conditions

Over the 15 week evaluation period a total of 100 939 valid observations by KSO-STREAMS was available. This corresponds to 28% of the astronomically possible observations (360 228). The remaining observations (72%) have been discarded due to exceedance of the accuracy requirements, detailed in Sect. 2.2. An overview about data availability (and

relative sunshine duration) per day during the evaluation period is given in Fig. 5. We note that only 43.7% of the theoretically possible sunshine duration was observed during the evaluation period, because of ambient weather conditions. KSO-STREAMS allows identifying the fraction of observations within manufacturer specified (i) active tracking accuracy during periods with direct solar radiation (DIR) exceeding 300 W m$^{-2}$ and (ii) passive tracking accuracy during periods with DIR below 300 W m$^{-2}$ . While the manufacturer specified tracking accuracy deteriorates during periods with DIR below

300 W m$^{-2}$ (passive tracking) this is not inherent to KSO-STREAMS observations. However, as the applied monitoring scheme is optical, it relies on visibility of the solar disk, thus no evaluation of the tracking accuracy is possible during periods of partial/full solar disk obstruction (see discussion below).

In the following we illustrate the performance of the sun-tracking device at Kanzelhöhe Observatory for four selected situations, illustrative for: (i) almost continuous clear-sky; (ii) clear sky interrupted by frontal movement; (iii) variable cloud

cover; and (iv) almost perpetual overcast conditions.

Almost continuous monitoring of the sun-tracking device pointing accuracy was possible on 7 May 2015, with prevailing clear sky conditions. Figure 6 shows the result of the zero point distance determined by KSO-STREAMS (in 15 second intervals) (panel (a)) as well as direct solar radiation (panel (b)); derived from ARAD (one minute averages) and the actual total output of the sun sensor of the sun-tracking device (in 10 minute increments) for this day. All available sun disk

centers, according to the selected restrictions (see Sect. 2.2), monitored on 7 May 2015 have been within the 0.1° limit, as specified in the BSRN guidelines. 90.7% of them have been within the manufacturer specified active tracking pointing accuracy of 0.02°. Individual sun disk centers deviate from the overall set, triggered by individual clouds affecting the determination of the sun limb. Nevertheless, also during these periods the pointing accuracy of the sun-tracking device lies well within manufacturer specifications for passive tracking (0.1°) and BSRN targets.

The 22 April 2015 is comparable with 7 May 2015, although clear sky conditions got interrupted through frontal movement between around 7:15 UT to 8:15 UT, indicated by the abrupt decline in DIR (Fig. 6d). No evaluation of the sun-tracking device pointing accuracy was possible during the frontal passage as thick cloud coverage affected solar limb detection. Before and after the frontal passage clear sky prevailed and KSO-STREAMS monitored pointing accuracies have been within BSRN targets and widely (89.2% of the observations) within manufacturer specifications for active tracking (see Fig.

6c).

Next we focus on the evaluation of sun-tracking accuracy during variable cloud cover as well as days with almost continuous cloud coverage, where active tracking mode (manufacturer requirement is denoted by a total output of the sun sensor of at least 300 W m$^{-2}$) and therefore its evaluation is only possible during short temporal increments.



April 12 and May 10, 2015 are representative for days with variable meteorological conditions, and therefore large variations in cloud cover. Periods with high (thick) cloud coverage, and limited direct radiation (Fig. 7 panels (b) and (d)), affect the ability of solar limb detection by KSO-STREAMS. During times with thinner clouds limb detection is possible as under clear sky conditions discussed above. Pointing accuracies are throughout within manufacturer specifications for passive

tracking (0.1°) and consequently also BSRN targets. However, despite DIR exceeding manufacturer requirements for active tracking frequently on these days only 68-78% of valid observations are within specifications for active tracking (Fig. 7 panel (a) and (c)).

Similar results are found on days with prevailing overcast conditions, where only small gaps in cloud cover occur. Figure 8 shows data on pointing accuracy and direct radiation on 20 May and 3 June 2015, which are representative for overcast days

during the evaluation period. On both days evaluation of the pointing accuracy of the sun-tracking device was only possible during small gaps in cloud cover. We evaluate tracking accuracy within manufacturer targets for active tracking on these days utilizing all observational data where the minimum of the ARAD direct radiation measurements (performed at a sample rate of 10 Hz) within a minute exceeds $300\ \mathrm{W\ m^{-2}}$. On 20 May 42.6% and on 3 June 2015 49.2% of these selected observations indeed fulfill the targeted accuracy of $\leq 0.02°$.

If the analysis is extended to the entire three month period we find that 96% of the observations (during periods where the minimum of the ARAD direct radiation measurements within a minute exceeds $300\ \mathrm{W\ m^{-2}}$) are within BSRN accuracy targets and about 75% within manufacturer specified active tracking mode limits. While the vast majority of observations during active tracking mode fulfills active tracking accuracy requirements difficulties arise during breaks in overcast conditions. Nevertheless it is important to note that even though larger fractions of observations on days with overcast

conditions do not fulfill active tracking requirements they still fulfill BSRN targets.

Finally we characterize the overall attainment of tracking accuracy within active tracking targets on days comprising the sets of days with (i) almost continuous clear-sky; (ii) clear-sky interrupted by frontal movement; (iii) variable cloud cover; and (iv) almost perpetual overcast conditions. On days with almost continuous clear-sky conditions and days with dominating clear-sky conditions interrupted by frontal movement BSRN targets are fully met and active tracking requirements are met

for 76.4% of valid observations. Active tracking accuracy requirements are less frequently met during variable cloud cover conditions (defined as days with $\leq 65\%$ and $> 15\%$ of theoretically possible daily KSO-STREAMS observations). During the three month period about half of the observational days (46 days) have been categorized as "variable" and 78.1% (99.9%) of observations fulfilled active tracking requirements (BSRN targets). Days with almost perpetual overcast conditions are defined as days with $\leq 15\%$ of theoretically possible KSO-STREAMS observations. On these days 64.3% of the available

valid observations fulfilled active tracking accuracy requirements and 99.5% BSRN targets. Calculated over all 85 days with available measurements during the evaluation period 72.9% of observations fulfill active tracking accuracy requirements and 99.8% fulfill BSRN targets. A detailed summary of achieved tracking accuracy (per category and total) is provided in Table 2.





## 4. Conclusions

Precise sun-tracking is necessary for high accuracy measurements of direct and diffuse solar radiation. Therefore rigid targets for sun-tracking pointing accuracies are specified in national and international radiation monitoring networks such as e.g., the Baseline Surface Radiation Network (BSRN) which specifies pointing accuracy requirements within 0.1°.

Sun-tracking devices fulfilling this pointing accuracy are available from a variety of instrument manufactures but none of the commercially available instruments comprises an automatic monitoring system allowing station operators independent evaluation of pointing accuracies during operation. In this manuscript we present KSO-STREAMS (KSO-SunTRackEr Accuracy Monitoring System), a platform independent, fully automated, and cost-effective system/method to evaluate the pointing accuracy of sun-tracking devices. During operation KSO-STREAMS is mounted like a pyrheliometer on the

sun-tracking device to ensure correct imaging of the Sun's position as identified by the tracking device.

To determine the pointing accuracy of the sun-tracking device operated at the Austrian RADiation Network (ARAD, Olefs et al. (2016)) site Kanzelhöhe Observatory (KSO) observations by KSO-STREAMS, taken over a 15 week period from March to June 2015 have been analyzed. Instrument performance is evaluated for four sets of ambient meteorological conditions: (i) almost continuous clear-sky; (ii) clear-sky interrupted by frontal movement; (iii) variable cloud cover; and (iv) almost

perpetual overcast conditions. The results show that 72.9% of all observations made during periods with more than 300 W m$^{-2}$ fulfill manufacturer specified active tracking accuracy requirements (0.02°) and 99.8% fulfill BSRN targets (0.1°). On days with almost continuous clear-sky conditions and/or days with clear-sky conditions interrupted by frontal movement BSRN requirements are fully satisfied and accuracies for active tracking are met for 76.4% of observations. Similar results are found for days with variable cloud-cover conditions. As expected, sun-tracking pointing accuracies are

lowest during days with almost perpetual overcast conditions; here 64.3% of observations meet active tracking requirements. Nevertheless BSRN accuracy targets are still almost completely met (99.5%) illustrating the strong performance of the sun-tracking system operated at KSO (SOLYS 2, Kipp & Zonen). We conclude that KSO-STREAMS provides valuable information on the quality of radiation measurement accuracies through evaluation of the underlying pointing accuracies of the operational sun-tracking device.

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



Table 1. Components of the KSO-STREAMS device and their characteristics

| Component | Characteristics |
|---|---|
| Circular aperture | Entrance window with a diameter of 12 mm |
| Filter block | Combination of neutral- and gold-coated bandpass-filter to prohibit detector saturation |
| Achromatic lens | Focal length of 60 mm |
| Camera | Compact network camera LAN; real-time images in VGA resolution; 1/4″ CMOS-sensor; adapted to power over Ethernet (PoE); special housing for outdoor usage |
| Housing and mounting system | Meet requirements of IP65 and is prepared to install the system like a pyrheliometer |



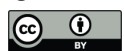

Table 2. Summary of achieved tracking accuracy for the determined sky-cover categories.

| Classification | Relative amount of valid observations (V) | Active tracking | BSRN requirement | # of valid observations | # of days |
|---|---|---|---|---|---|
| Almost continuous clear-sky or clear-sky interrupted by frontal movement | V > 65% | 76.4% | 100% | 33 179 | 14 |
| Variable cloud cover | 15% < V ≤ 65% | 78.1% | 99.9% | 62 735 | 46 |
| Almost perpetual overcast conditions | V ≤ 15% | 64.3% | 99.5% | 5 025 | 25 |
| Total | | 72.9% | 99.8% | 100 939 | 85 |



Figure 1. Instrumental setup: **(a)** operating device; **(b)** optical layout (circular aperture (A), filter block (F), achromatic lens (L), and camera chip (C)); **(c)** mounting system; **(d)** radiation platform with KSO-STREAMS mounted on the sun-tracking device (SOLYS 2; Kipp & Zonen) at ARAD site Kanzelhöhe Observatory.





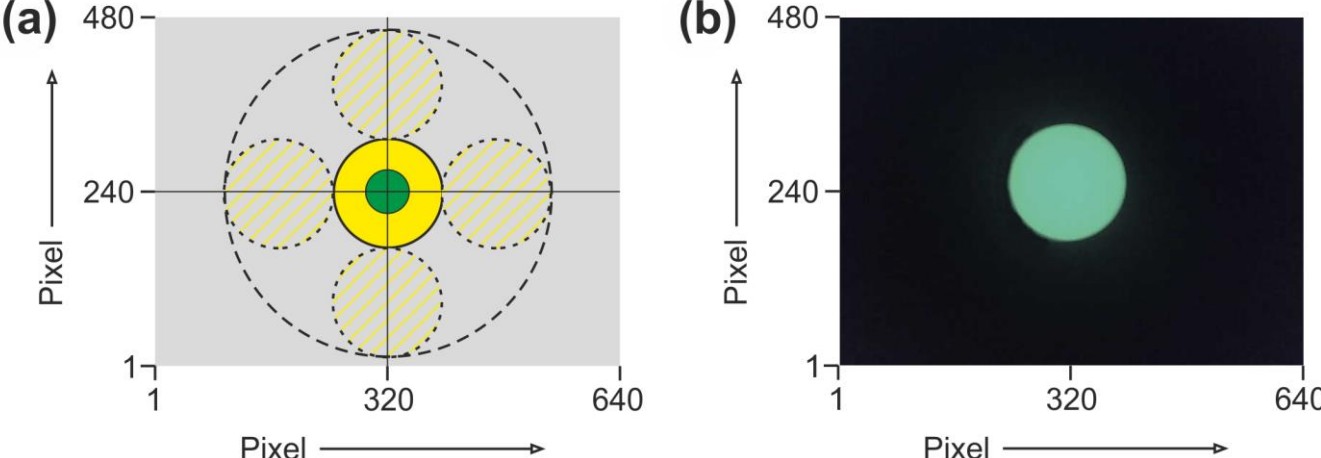

Figure 2. Solar disk images on the sensor array (VGA resolution) of the compact network camera of KSO-STREAMS. **(a)** range of the detectable free movement of the positions of solar disk images within the detector array due to possible misalignment of the sun-tracking device (yellow area: optimal position of solar disk image; green marked zone: possible center of the solar disk image to be within BSRN requirements; yellow shaded: detectable misalignment of the sun-tracking device through KSO-STREAMS); **(b)** typical solar disk image under clear-sky conditions





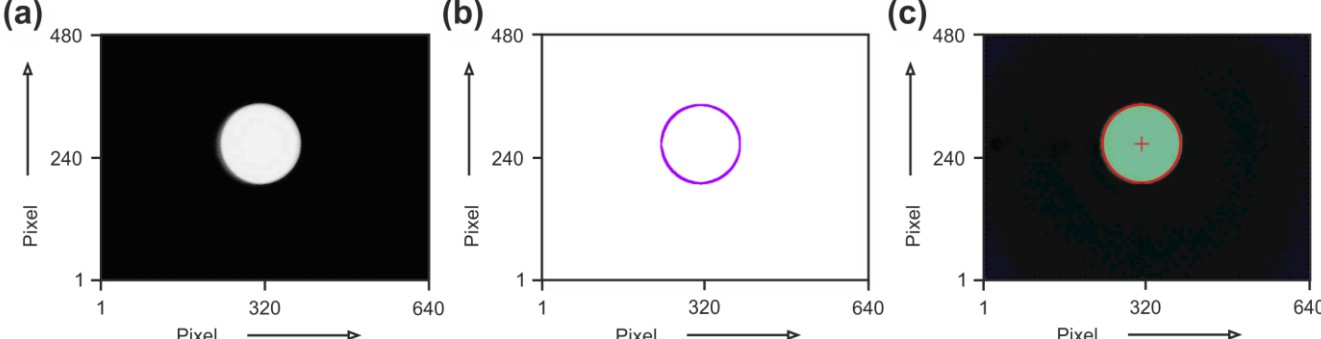

Figure 3. Illustration of individual steps in KSO-STREAMS image processing to derive a circular fit to solar limb pixels: **(a)** greyscale transformed solar disk image (original KSO-STREAMS picture is shown as background in **(c)**); **(b)** solar limb

5   pixels (purple) identified through application of a Sobel operator to **(a)**; **(c)** best fitting circle (red) to solar limb pixels from **(b)** derived by a least square fitting approach superimposed on the original KSO-STREAMS picture. The red cross in **(c)** marks the center of the fitted circle.





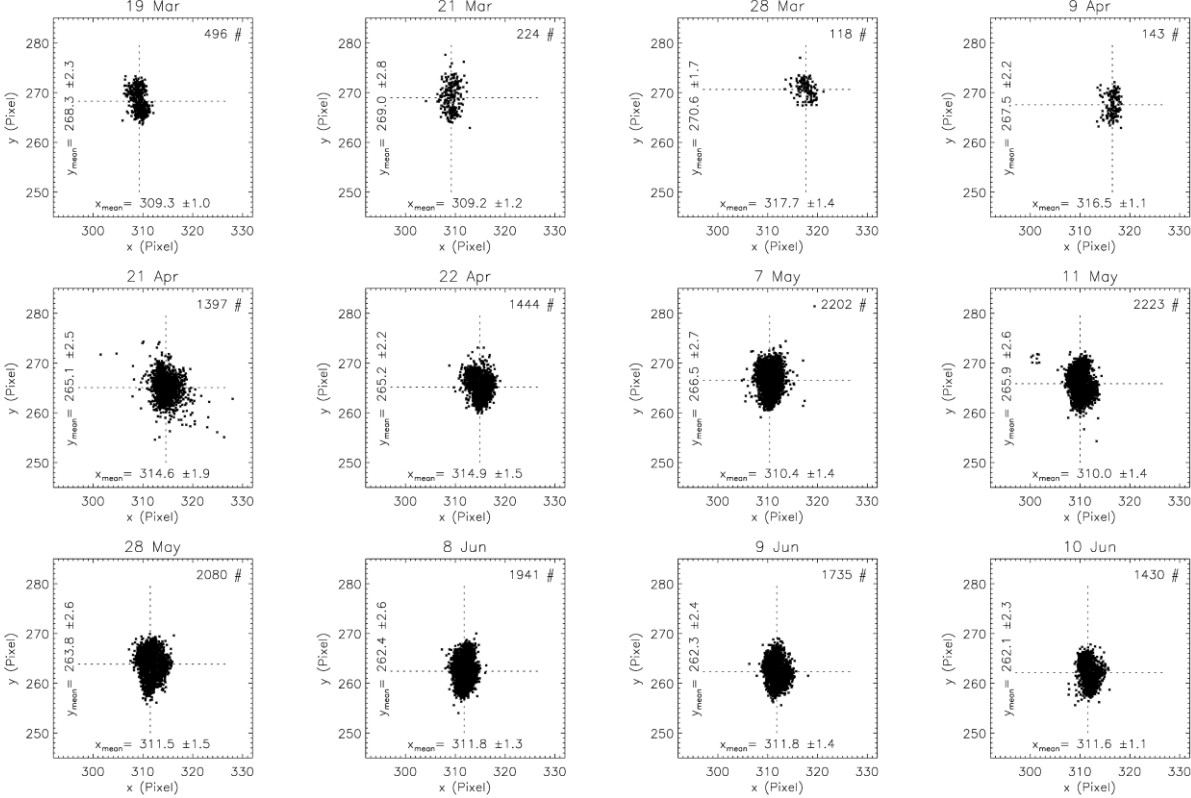

Figure 4. Solar disk center positions (black dots) determined by KSO-STREAMS, used for the determination of the mean daily zero point centers on 12 clear-sky days in 2015. The zero point center of KSO-STREAMS is defined as the mean value of these 12 zero point centers.





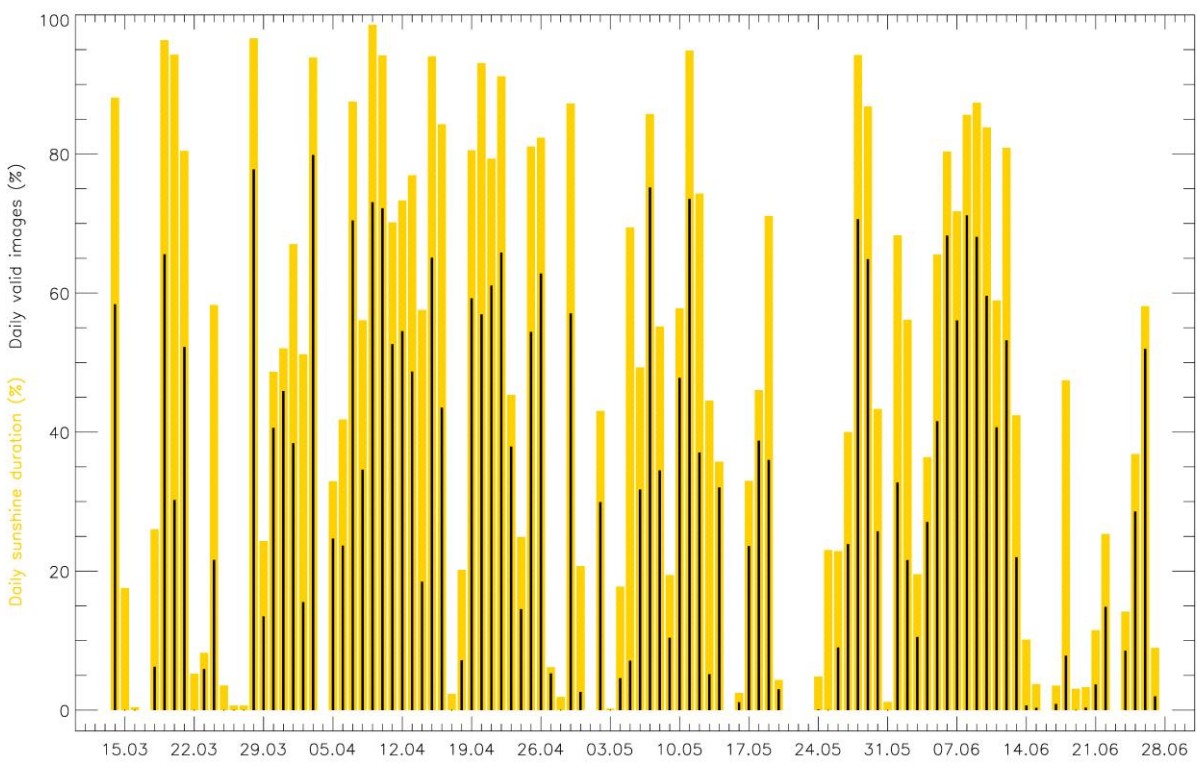

Figure 5. Relative daily sunshine duration (yellow) and valid KSO-STREAMS results (black) during the evaluation period from 14 March to 27 June 2015.




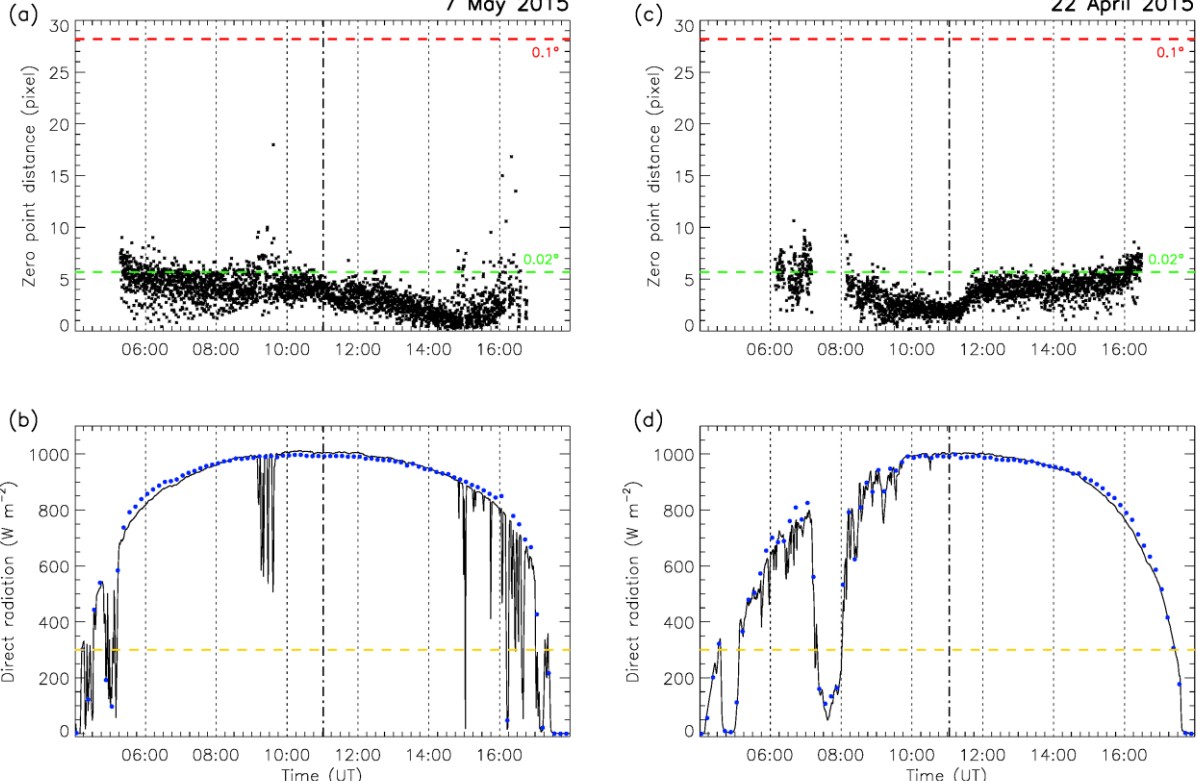

Figure 6. Performance of the sun-tracking device at ARAD site Kanzelhöhe Observatory for days with almost continuous clear-sky (7 May 2015; left column) and clear-sky interrupted by frontal movement (22 April 2015; right column): **(a)** radial distance from the zero point of each valid image (asterisks); **(b)** direct radiation (1 min averages) derived from ARAD site Kanzelhöhe Observatory (line) and actual sun sensor measurements of the sun-tracking device (in 10 min intervals, blue dots) on 7 May 2015. **(c)-(d)** as **(a)-(b)** but for 22 April 2015. Limits of active tracking (i.e. alignment within 0.02° accuracy – green line) and passive tracking (i.e. alignment within 0.1° accuracy – red line) are shown in **(a)** and **(c).** The yellow dashed line in panels **(b)** and **(d)** indicates the manufacturer specified minimum of direct radiation (300 W m$^{-2}$) needed for active tracking mode. The vertical black dotdashed line indicates astronomical noon in all panels.





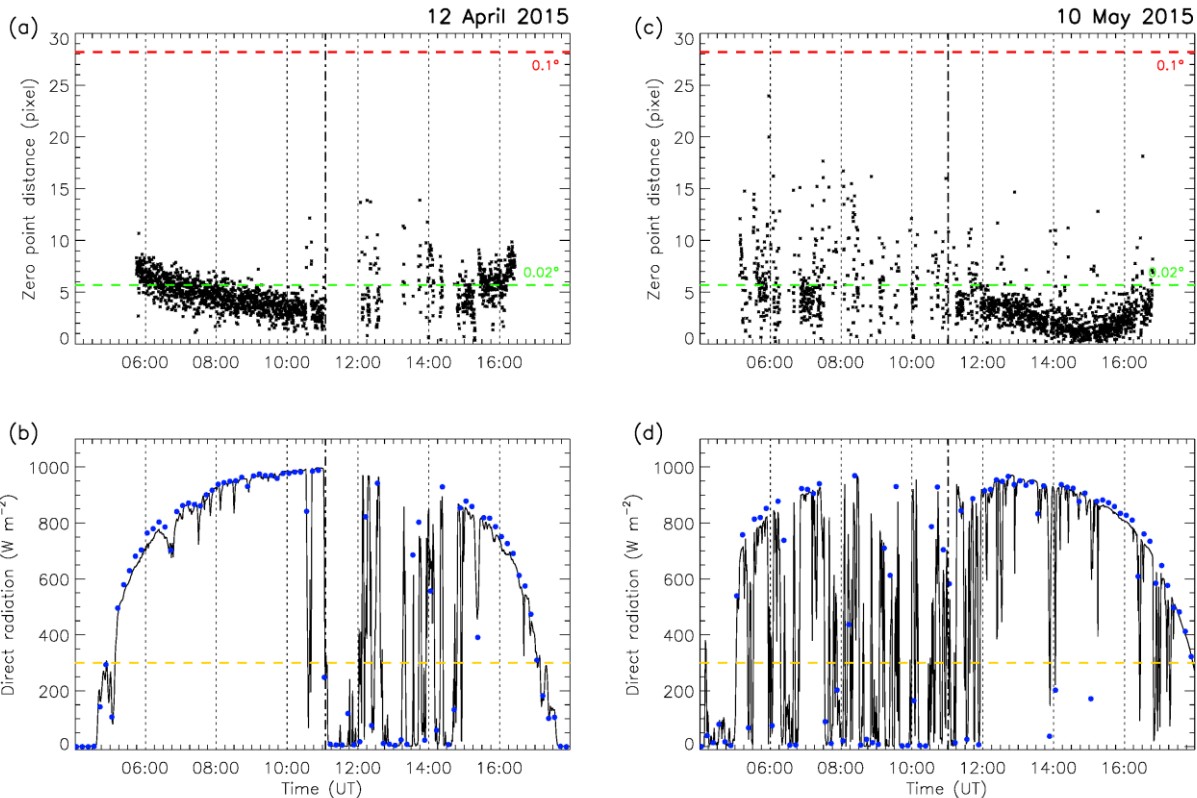

Figure 7. As Fig. 6 but for days with variable cloud cover (left: 12 April 2015; right: 10 May 2015).





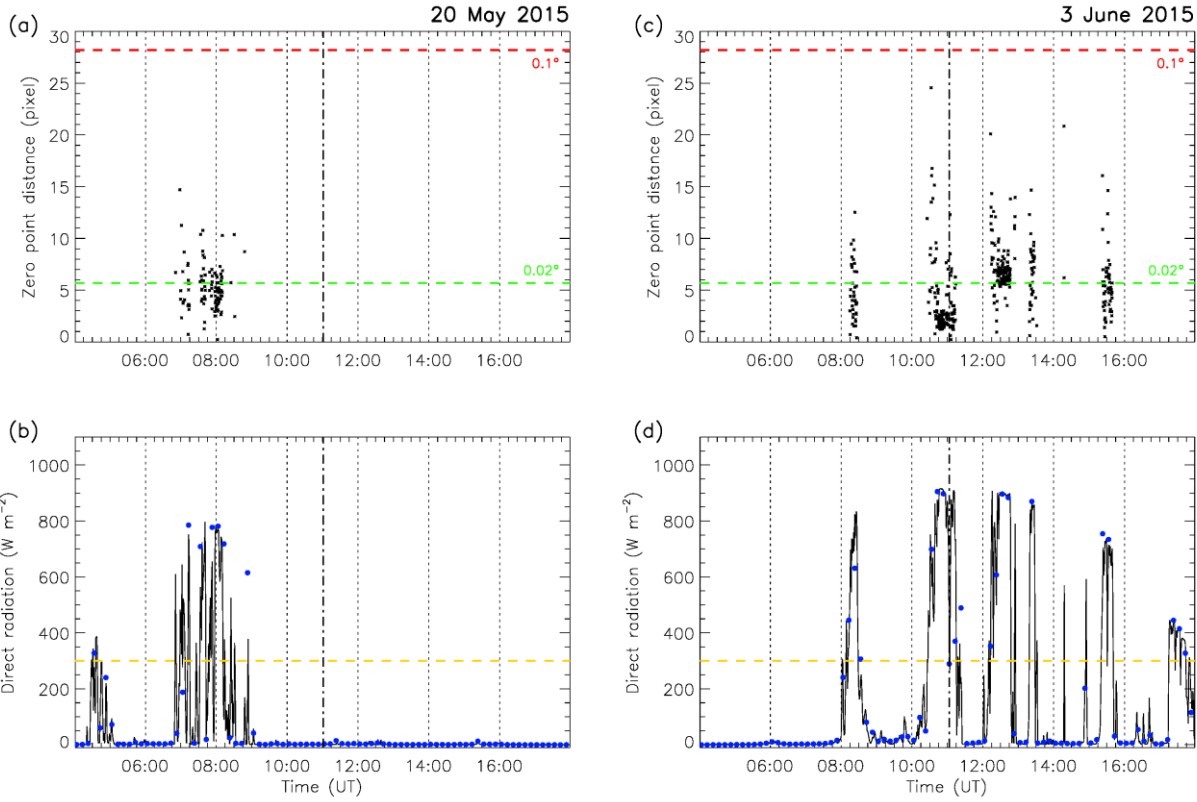

Figure 8. As Fig. 6 but for days with almost perpetual overcast conditions (left: 20 May 2015; right: 3 June 2015).

