# Peer review of "An automated method for the evaluation of the pointing accuracy of sun-tracking devices"

_Atmospheric Measurement Techniques, 2016_

## Referee Comment (RC1) · Anonymous Referee #1 · 4 Oct 2016

This paper introduces a method for the assessment of pointing accuracy in solar radiation measurements as carried out e.g. in the Baseline Surface Radiation Network (BSRN). The method relies on a camera system mounted similar to a pyrheliometer that allows continuous monitoring of the position of the solar disc and thereby evaluates the sun tracking accuracy. After a description of the instrumental setup and the data analysis scheme, the paper provides an evaluation of the system performance based on several months of field measurements.

General Comments

While somewhat similar approaches to the evaluation of pointing accuracy have already been proposed and implemented for other sun-tracking instruments (see specific

comments), the setup presented in this paper provides a simple and widely applicable solution to detect measurements with non-optimum pointing accuracy in solar radiation monitoring instruments which need to be identified to avoid inaccurate measurements. The proposed scheme is therefore suitable for implementation in many other measurement sites. The paper contains a clear and concise description of this method and should therefore be considered for publication in AMT after a number of minor comments have been adressed.

Specific Comments

Page 2, line 25: Since this is the key motivator for the paper, a more detailed outline of why pointing accuracy is an important requirement for accurate solar radiation measurements would be useful for the reader (see e.g. McArthur, 2005, Annex D).

Page 2, line 30: While pointing accuracy is not commonly monitored automatically for the instruments used by the authors, this is a common challenge for most types of sun-pointing instruments. Therefore, several previous efforts have been made to implement sytems that monitor this parameter for other instruments. Since these methods are partly similar to the concept proposed here, a short discussion of relevant previous work would be appropriate. As an example, in the field of solar FTIR spectrometry, a camera-based system partly similar to the approach shown in this study has been described by Gisi et al. (2011). An alternative approach for the evaluation of sun pointing accuracy which does not require a camera setup has been presented by Reichert et al. (2015). Other equally relevant studies possibly exist for different types of sun-tracking instruments.

Page 5, Sect. 2.3: If I understand correctly, this initial zero point determination method does not allow to distinguish between systematic mispointing of the sun tracker used in the solar radiation measurements and a misaligment of the KSO-STREAMS system relative to the sun tracker's line of sight. Can you elaborate if a significant systematic mispointing of the sun tracker is possible and, if yes, how such an effect will influence

the accuracy of the solar radiation measurements?

Page 5, line 21: "mainly affected by different atmospheric conditions" - can you specify more explicitly which atmospheric parameters influence the zero point position?

Page 8, lines 15-22: You conclude that, as expected, the quadrant sensor-based tracking is less accurate for measurement days with cloud influence. Can you discuss in more detail why you are confident that this finding is solely due to the accuracy of the tracking device and an influence by a possible dependence of the KSO-STREAMS analysis algorithm's performance on the presence of clouds (e.g. for the solar limb detection) can be excluded?

References

Gisi, M., Hase, F., Dohe, S., and Blumenstock, T.: Camtracker: a new camera controlled high precision solar tracker system for FTIR-spectrometers, Atmos. Meas. Tech., 4, 47–54, doi:10.5194/amt-4-47-2011, 2011.

McArthur, L. J. B.: Baseline Surface Radiation Network (BSRN) Operations Manual Version 2.1WCRP-121, WMO/TD-No. 1274, 2005.

Reichert, A., Hausmann, P., and Sussmann, R.: Pointing errors in solar absorption spectrometry – correction scheme and its validation, Atmos. Meas. Tech., 8, 3715-3728, doi:10.5194/amt-8-3715-2015, 2015.

---

## Author Comment (AC1) · 25 Oct 2016

**Response to the comments of Referee #1:**

We thank the referee for the positive judgment of our manuscript and the useful comments provided in the report. Below we provide the individual referee comments and *our response in italics*.

Page 2, line 25: Since this is the key motivator for the paper, a more detailed outline of why pointing accuracy is an important requirement for accurate solar radiation measurements would be useful for the reader (see e.g. McArthur, 2005, Annex D).

*We thank the referee for addressing this point and in particular the study of G. Major presented in Annex D of McArthur (2005).*

*Correct alignment of the sun-tracking device is crucial for high accuracy monitoring of diffuse (DIF) and direct (DIR) solar radiation. Precise alignment is of highest priority for the monitoring of DIF as already small misalignments can significantly affect the measurement accuracy. For measurements of DIF one strives to solely shade the pyranometer's glass dome, to mask as little of the diffuse component as possible while simultaneously shielding direct solar irradiance. BSRN strives to achieve measurements at highest possible accuracy and defines in its operations manual (McArthur, 2005) multiple times a pointing accuracy requirement < 0.1° (Pages 11, 54).*

*For the monitoring of DIR pointing accuracy is also important, but maybe less crucial as for DIF. This is also addressed in the study of G. Major presented in Annex D of McArthur (2005). This study concludes that pointing errors are (i) neglible if smaller than a pyrheliometer's slope angle; but (ii) increasingly important with increasing error as measured irradiance decreases rapidly with increasing mispointing. Particularly the contribution of the circumsolar sky to the irradiance measured by pyrheliometer depends strongly on three factors: atmospheric aerosol loading, solar zenith angle and pointing error (see Fig. D1.4 in McArthur, 2005).*

*We will expand the corresponding section discussing the importance of pointing accuracy in solar radiation monitoring by including the information provided above in the revised manuscript.*

Page 2, line 30: While pointing accuracy is not commonly monitored automatically for the instruments used by the authors, this is a common challenge for most types of sun-pointing instruments. Therefore, several previous efforts have been made to implement systems that monitor this parameter for other instruments. Since these methods are partly similar to the concept proposed here, a short discussion of relevant previous work would be appropriate. As an example, in the field of solar FTIR spectrometry, a camera-based system partly similar to the approach shown in this study has been described by Gisi et al. (2011). An alternative approach for the evaluation of sun pointing accuracy which does not require a camera setup has been presented by Reichert et al. (2015). Other equally relevant studies possibly exist for different types of sun-tracking instruments.

*In the original manuscript we aimed on focusing the discussion solely on sun-tracking in solar radiation monitoring. We are aware that similar problems arise in other fields and that innovative camera-based and camera-free approaches have been implemented in the field of FTIR measurements. In the context of the available solar radiation monitoring instrumentation no camera-free observing option was straightforward implementable. KSO-STREAMS is therefore, as also acknowledge by the referee, intended as simple and widely applicable solution at many measurement sites. To provide further context to the challenge arising in the determination of pointing accuracy and an overview of alternative tools/methods developed in various fields we will extend the corresponding section in the introduction of the manuscript.*

Page 5, Sect. 2.3: If I understand correctly, this initial zero point determination method does not allow to distinguish between systematic mispointing of the sun tracker used in the solar radiation measurements and a misaligment of the KSO-STREAMS system relative to the sun tracker's line of sight. Can you elaborate if a significant systematic mispointing of the sun tracker is possible and, if yes, how such an effect will influence the accuracy of the solar radiation measurements?

*The reviewer is correct with this assessment; however significant misalignment of the sun-tracking device can be excluded by following the step wise installation procedure described in the manufacturer manual (http://www.kippzonen.com/Product/20/SOLYS2-Sun-Tracker).*

*The set up procedure includes double alignment control (with and without a pyrheliometer) before the sun-sensor is mounted on the sun-tracking device. If the control procedures provided in the manual are not followed correctly a misalignment would/could arise, which would affect the accuracy of the radiation measurements.*

Page 5, line 21: "mainly affected by different atmospheric conditions" - can you specify more explicitly which atmospheric parameters influence the zero point position?

*The most important atmospheric parameters influencing zero point determination are turbidity and humidity. Clouds would be important – and to avoid cloud effects only clear-sky periods have been included in zero-point determination.*

Page 8, lines 15-22: You conclude that, as expected, the quadrant sensor-based tracking is less accurate for measurement days with cloud influence. Can you discuss in more detail why you are confident that this finding is solely due to the accuracy of the tracking device and an influence by a possible dependence of the KSO-STREAMS analysis algorithm's performance on the presence of clouds (e.g. for the solar limb detection) can be excluded?

*We apologize for not being clear enough in this section. Tracking accuracies are only documented for valid KSO-STREAMS observations, i.e. observations where the solar limb is within the specified limit (+5% of the largest imaged Sun radius and -5% of the smallest possible Sun radius throughout the year) and the uncertainty of the circle fit is one pixel or less. In all three classes of observations analyzed (clear-sky, variable cloud cover, almost perpetual overcast conditions) analysis is restricted to valid KSO-STREAMS observations.*

*On days with almost perpetual cloud cover 15% or less of observations are fulfilling these criteria (see Table 2 in the discussion paper), while other observations do not meet one or both accuracy criteria due to cloud effects; here particularly limb detection is influenced by masking of thick clouds.*

*Of those observations fulfilling the specified accuracy criteria on days with almost perpetual overcast conditions only 64.3% meet active tracking requirements which is substantially less than for the other categories (i.e., days with valid KSO-STREAMS observations > 15%). We will rephrase this section for clarity in the revised version of the manuscript and apologize for any confusion.*

*References*

*McArthur, L. J. B.: Baseline Surface Radiation Network (BSRN) Operations Manual Version 2.1WCRP-121, WMO/TD-No. 1274, 2005.*

---

## Referee Comment (RC2) · Dietmar J. Baumgartner et al. · 1 Feb 2017

This paper introduces s device that can be used to assess the accuracy of solar tracking systems, which is based on processing of solar images recorded by a camera mounted on the tracking system. It is a useful tool which would help station operators to ensure the quality of pyrheliometer of diffuse radiation measurements obtained by shadowing disks or spheres. Furthermore, results of the system's performance at a monitoring site where direct irradiance measurements are conducted is presented. The paper's contents fall well within the scopes of AMT. The presentation and methodology are clear, apart from some minor issues discussed below, therefore I believe that he paper can accepted for publication after minor revisions.

Specific comments

P3, L14: Please clarify what is meant by detecting misalignment of the solar disk of 0.5°, while the system is designed to have much better precision.

P3, L22: it is not clear to me what is meant by "mounting in perfect horizontal alignment on the tracking system". Consequently, I cannot understand the need for derotating the images.

P3, L26: In step (iv) what kind of data are used for the fitted line? Is it the position of the center vs time, vs azimuth angle, or what? Furthermore, it is not clear how the angle ($\varphi$) is defined, since -to my understanding- the (solar) image border is circular. Maybe a sketch would help the reader to understand more easily the geometry.

P5, L3: It is stated that the error in the circle fit is one pixel. Doesn't this error depend on the resolution of the camera? Please state in the text how many pixels (on average) are along the diameter of the solar image.

P5, L15: Please explain what exactly is the "zero point position". It is introduced for the first time here, without having been defined. I would guess that it is a kind of an offset due a misalignment of the system on the tracker. However, later in the discussion of Figure 6 I see that the zero point is used to assess the tracking accuracy of a commercial tracker, which confuses me.
* * *